# Seasonal variations in tuberculosis diagnosis among HIV-positive individuals in Southern Africa: analysis of cohort studies at antiretroviral treatment programmes

Marie Ballif,[1] Kathrin Zürcher,[1] Stewart E Reid,[2,3] Andrew Boulle,[4,5] Matthew P Fox,[6,7] Hans W Prozesky,[8] Cleophas Chimbetete,[9] Marcel Zwahlen,[1] Matthias Egger,[1,4] Lukas Fenner,[1] On behalf of the International Epidemiology Database to Evaluate AIDS in Southern Africa (IeDEA-SA)

For numbered affiliations see end of article.

**Correspondence to**
Dr Marie Ballif;
marie.ballif@ispm.unibe.ch

## ABSTRACT

**Objectives** Seasonal variations in tuberculosis diagnoses have been attributed to seasonal climatic changes and indoor crowding during colder winter months. We investigated trends in pulmonary tuberculosis (PTB) diagnosis at antiretroviral therapy (ART) programmes in Southern Africa.

**Setting** Five ART programmes participating in the International Epidemiology Database to Evaluate AIDS in South Africa, Zambia and Zimbabwe.

**Participants** We analysed data of 331 634 HIV-positive adults (>15 years), who initiated ART between January 2004 and December 2014.

**Primary outcome measure** We calculated aggregated averages in monthly counts of PTB diagnoses and ART initiations. To account for time trends, we compared deviations of monthly event counts to yearly averages, and calculated correlation coefficients. We used multivariable regressions to assess associations between deviations of monthly ART initiation and PTB diagnosis counts from yearly averages, adjusted for monthly air temperatures and geographical latitude. As controls, we used Kaposi sarcoma and extrapulmonary tuberculosis (EPTB) diagnoses.

**Results** All programmes showed monthly variations in PTB diagnoses that paralleled fluctuations in ART initiations, with recurrent patterns across 2004–2014. The strongest drops in PTB diagnoses occurred in December, followed by April–May in Zimbabwe and South Africa. This corresponded to holiday seasons, when clinical activities are reduced. We observed little monthly variation in ART initiations and PTB diagnoses in Zambia. Correlation coefficients supported parallel trends in ART initiations and PTB diagnoses (correlation coefficient: 0.28, 95% CI 0.21 to 0.35, P<0.001). Monthly temperatures and latitude did not substantially change regression coefficients between ART initiations and PTB diagnoses. Trends in Kaposi sarcoma and EPTB diagnoses similarly followed changes in ART initiations throughout the year.

**Conclusions** Monthly variations in PTB diagnosis at ART programmes in Southern Africa likely occurred regardless

## Strengths and limitations of this study

► We analysed a large dataset from HIV-positive patients routinely collected over 10 years at antiretroviral therapy (ART) programmes in low-income and middle-income countries from the Southern African region.
► Included ART programmes represent a wide range of countries with different climatic seasons according to their geographical latitude.
► We compared trends in pulmonary tuberculosis diagnosis to trends in Kaposi sarcoma and extrapulmonary tuberculosis diagnoses, for which no effect due to seasonal climatic changes was expected.
► Our study is limited by the lack of social and environmental information on participants (socioeconomic factors and housing conditions).
► Trends in tuberculosis diagnosis and other opportunistic infections in HIV-positive individuals may differ from trends in HIV-negative populations.

of seasonal variations in temperatures or latitude and reflected fluctuations in clinical activities and changes in health-seeking behaviour throughout the year, rather than climatic factors.

## INTRODUCTION

Countries in different regions of the world have reported seasonal fluctuations in tuberculosis (TB) cases.[1–6] In temperate climate zones, peaks in TB diagnosis are generally observed in the spring, following the colder times of the year. The reasons for these variations are multiple, with both social and environmental factors likely playing a role.[7] Peaks in TB diagnosis following winter time have been associated with several factors: more

**Table 1** Characteristics of participating ART programmes

| Characteristics | South Africa, Khayelitsha | South Africa, Themba Lethu | South Africa, Tygerberg | Zambia, CIDRZ | Zimbabwe, Newlands |
|---|---|---|---|---|---|
| Geographical latitude (°) | −34.05 | −33.91 | −26.12 | −15.41 | −17.81 |
| Setting | Urban | Urban | Urban | Combination of urban and periurban health centres | Urban |
| Level of care | Primary | Secondary | Tertiary | Primary | Primary |
| TB case definition | Majority microbiologically confirmed, others based on clinical presentation | Microbiologically confirmed | Majority microbiologically confirmed, others based on clinical presentation | Majority microbiologically confirmed, others based on clinical presentation | Majority microbiologically confirmed, others based on clinical presentation |
| Cost of TB diagnosis to patients | Available at no cost | Available at no cost | Partial payment | Available at no cost or partial payment (depending on facility) | Available at no cost |
| Cotrimoxazole for prevention | All patients | According to CD4 cell count | According to CD4 cell count | According to CD4 cell count or to all patients (depending on facility) | According to CD4 cell count |
| Integration of HIV/TB clinical services | Same facility/same day appointments | Same facility/same day appointments | Cross-referral | Cross referral | Same facility/same day appointments |
| TB screening at ART enrolment | All patients, symptomatic screening and diagnostic tests | All patients, symptomatic screening only | All patients, symptomatic screening only | All patients, symptomatic screening and diagnostic tests at selected sites | All patients, symptomatic screening only |

Data sources: ref 15.
ART, antiretroviral therapy; CIDRZ, Centre for Infectious Disease Research in Zambia; TB, tuberculosis.

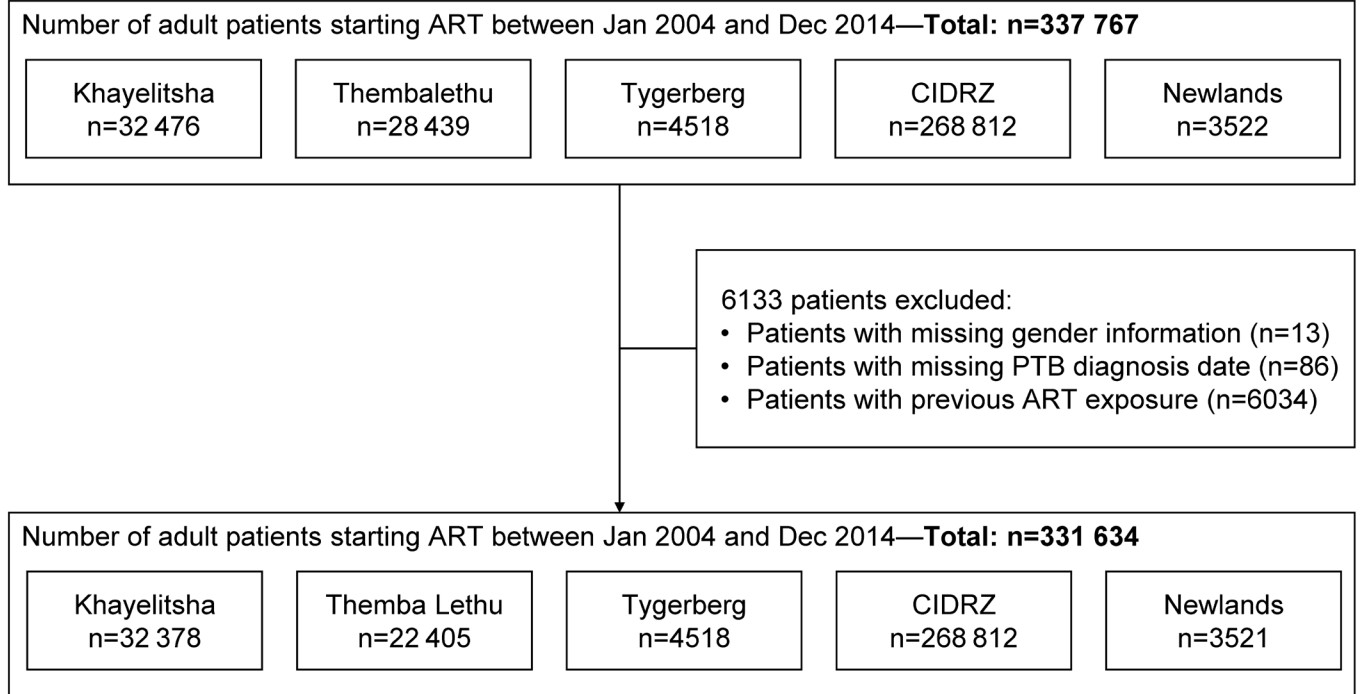

**Figure 1** Selection of patients for the analysis. ART, antiretroviral therapy; CIDRZ, Centre for Infectious Disease Research in Zambia; PTB, pulmonary tuberculosis.

**Table 2** Characteristics of patients diagnosed with PTB between January 2004 and December 2014 compared with others

| Characteristics | Patients diagnosed with PTB | Other patients | All |
|---|---|---|---|
| **Age at ART start (median, IQR), years** | 35 (30–41) | 34 (29–41) | 34 (29–41) |
| *Missing, n (%)* | *0* | *0* | *0* |
| **Gender, n (%)** | | | |
| Woman | 25 227 (51.1) | 181 847 (64.4) | 207 074 (62.4) |
| Man | 24 140 (48.9) | 100 420 (35.6) | 124 560 (37.6) |
| *Missing, n (%)* | *0* | *0* | *0* |
| **CD4 cell count at ART start (median, IQR), cells/µL** | 116 (52–200) | 166 (87–254) | 158 (80–246) |
| *Missing, n (%)* | *24 237 (48.8)* | *150 058 (53.2)* | *174 295 (52.6)* |
| **WHO clinical stage, n (%)** | | | |
| Stages 1 and 2 | 4916 (9.9) | 157 322 (55.7) | 162 238 (48.9) |
| Stages 3 and 4 | 41 744 (84.6) | 106 836 (37.9) | 148 580 (44.8) |
| Unknown | 2707 (5.5) | 18 053 (6.4) | 20 816 (6.3) |
| *Missing, n (%)* | *0* | *56 (0.02)* | *56 (0.02)* |
| **Treatment programme, n (%)** | | | |
| South Africa, Khayelitsha | 7357 (14.9) | 25 021 (8.9) | 32 378 (9.8) |
| South Africa, Themba Lethu | 5438 (11.0) | 16 967 (6.0) | 22 405 (6.8) |
| South Africa, Tygerberg | 861 (1.7) | 3657 (1.3) | 4518 (1.4) |
| Zambia, CIDRZ | 35 132 (71.2) | 233 680 (82.8) | 268 812 (81.1) |
| Zimbabwe, Newlands | 579 (1.2) | 2942 (1.0) | 3521 (1.1) |
| *Missing, n (%)* | *0* | *0* | *0* |
| **Total, n (%)** | **49 367 (14.9)** | **282 267 (85.1)** | **331 634 (100)** |

Missing values are indicated in italic.

ART, antiretroviral therapy; CIDRZ, Centre for Infectious Disease Research in Zambia; PTB, pulmonary tuberculosis.

time spent indoor in poorly ventilated rooms, which increases the risk of TB transmission; poorer access to healthcare when rough weather conditions make transport difficult; and lower vitamin D levels due to reduced sunlight exposure, which impairs adequate immune response against infections.[1 3 5 8–10]

In 2015, an estimated 1.2 million of people living with HIV developed TB, and one-third of them died from TB, making TB the leading cause of death in this population.[11–13] However, little is known about the seasonal variation of TB diagnoses among HIV-positive individuals in care at antiretroviral therapy (ART) programmes in sub-Saharan Africa, where the burden of HIV/TB coinfection is the highest.[13] We hypothesised that fluctuations in TB diagnoses throughout the year would likely be stronger in countries located farther away from the equator and that experience more pronounced climatic changes, as opposed to countries located closer to the equator. We investigated seasonal trends in pulmonary TB (PTB) diagnosis at five ART programmes participating in the International Epidemiology Databases to Evaluate AIDS collaboration in Southern Africa (IeDEA-SA). These ART programmes are located at different geographical latitudes in South Africa, Zambia and Zimbabwe and present differently marked climatic seasons.

## METHODS

We included all HIV-positive persons aged ≥16 years who enrolled between 1 January 2004 and 31 December 2014 in five ART programmes that participate in the IeDEA-SA collaboration (www.iedea-sa.org) and systematically collect information on opportunistic infections (OIs)[14]: Khayelitsha ART programme, Themba Lethu clinic and Tygerberg Academic Hospital in South Africa; Centre for Infectious Disease Research in Zambia (CIDRZ); and Newlands Clinic in Zimbabwe. IeDEA-SA sites provide data collected as part of routine HIV care, including TB diagnosis, at enrolment in the ART programme and at each follow-up visit. Site characteristics are shown in table 1,[15] and the patient selection in figure 1.

All study sites have approval from a local institutional review board or ethics committee to collect data and participate in IeDEA-SA projects.

PTB was defined according to the case definition in use at the participating IeDEA-SA sites (table 1).[15] We restricted the main analysis to PTB, because the diagnosis of PTB has been associated with seasonal climatic variations.[1 2 7] For comparison, we analysed monthly counts of patients diagnosed with extrapulmonary tuberculosis (EPTB) and the AIDS-defining cancer Kaposi sarcoma (KS), which are not expected to be seasonal. One record

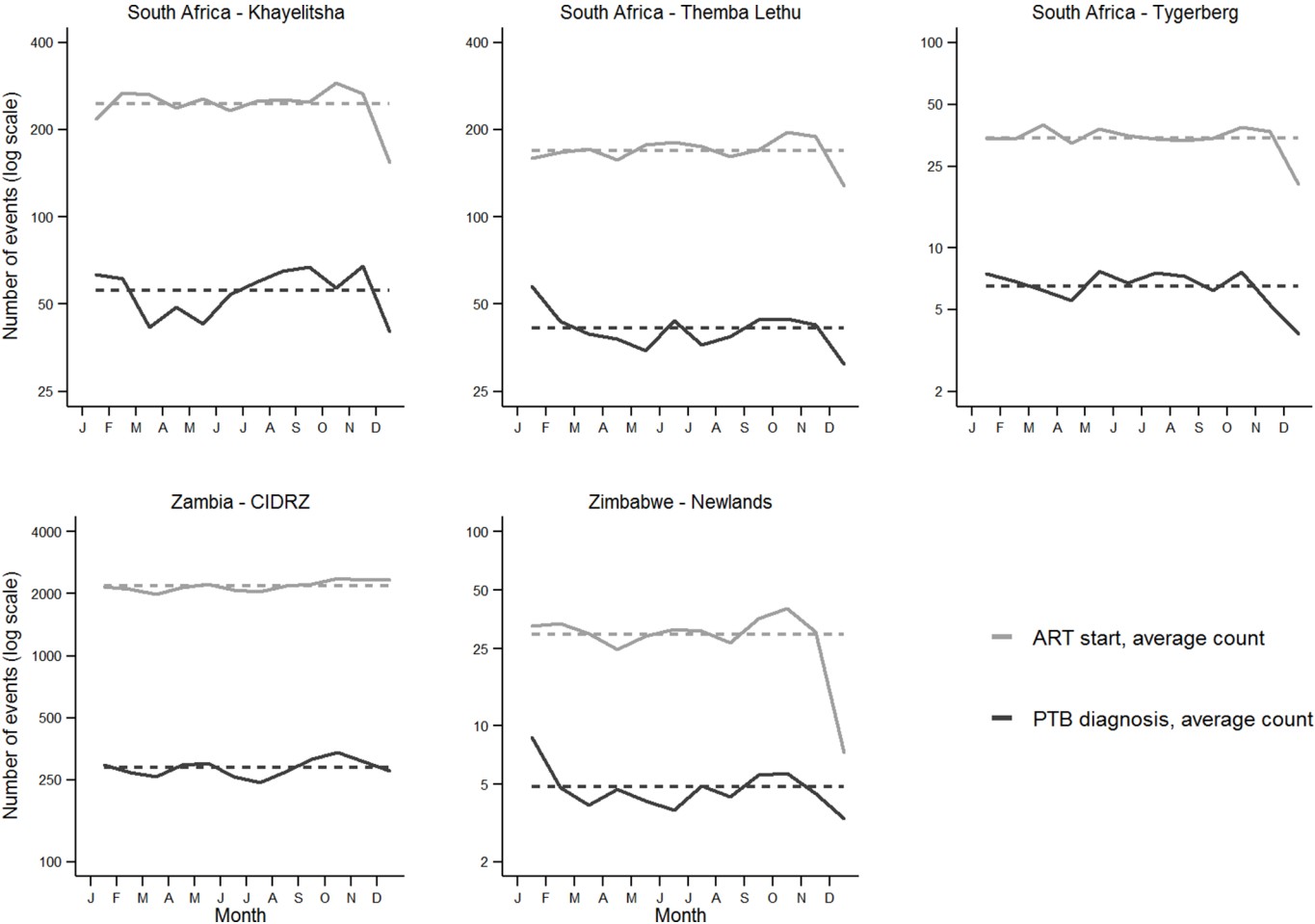

**Figure 2** Aggregated averages in monthly counts of patients newly started on ART and monthly counts of PTB diagnosis (aggregate averages 2004–2014, logarithmic scale). Dashed horizontal lines indicate overall monthly means. ART, antiretroviral therapy; PTB, pulmonary tuberculosis.

per patient was analysed. If a patient had several episodes of the same disease (PTB, EPTB or KS), we only considered the first event of each. If a patient was diagnosed with multiple diseases of interest, priority was given to the record with PTB diagnosis, followed by EPTB and KS. We obtained mean monthly air temperatures (°C) based on 1961–1990 averages at each location.[16–19]

For each site, we calculated aggregated averages as the average number of events per calendar month for 2004–2014. We accounted for calendar time trends over the study period by comparing deviations of monthly event counts to yearly averages. Site-specific aggregated averages and monthly deviations were calculated for ART enrolments and for PTB, EPTB and KS diagnoses, overall or stratified by gender and age. We assessed the association between monthly deviations in the number of ART enrolments and the number of PTB diagnoses by calculating pairwise Pearson's correlation coefficients.[20] We used Fisher's transformation to calculate 95% CI for the correlation coefficient. Finally, we used linear regression models to assess associations between the monthly deviations of PTB, EPTB or KS diagnosis counts and the monthly deviations of ART initiation counts, unadjusted and adjusted for monthly deviations of air temperatures

from yearly averages and for the geographical latitude of the ART programmes' locations.

All analyses were performed in Stata V.14.1 (Stata Corporation, College Station, Texas, USA).

### RESULTS
We analysed data of 331 634 HIV-positive adults (figure 1). Of these, 49 367 (14.9%) were diagnosed with PTB. The median follow-up time was 2.0 years (IQR, 0.7–4.2), patients' median age was 34 years (IQR, 29–41) and 207 074 (62.4%) were women (table 2). Median CD4 cell count at ART initiation of these patients was 116 cells/mm$^3$ (IQR, 52–200) in comparison with a median cell count of 166 (IQR, 87–254) in patients who were never diagnosed with PTB.

The mean number of PTB diagnoses per month was 55.7 (range 40.2–67.8) in Khayelitsha, 41.2 (31.1–57.6) in Themba Lethu, 6.5 (3.8–7.7) in Tygerberg, 285.6 (243.2–338.2) in CIDRZ and 4.8 (3.3–8.7) in Newlands (figure 2). At each site, monthly variations in numbers of PTB diagnoses were parallel to trends in ART enrolments. Aggregated averages over the study period showed that monthly fluctuations in PTB diagnoses were present at all

**Table 3** Correlation coefficients between deviations of monthly counts of ART enrolments from the yearly average and deviations of monthly counts of PTB, EPTB and KS diagnosis from the yearly averages

| ART programme | ART enrolments Median (IQR) | PTB diagnoses Median (IQR) | Correlation (95% CI) | P value | EPTB diagnoses Median (IQR) | Correlation (95% CI) | P value | KS diagnoses Median (IQR) | Correlation (95% CI) | P value |
|---|---|---|---|---|---|---|---|---|---|---|
| South Africa, Khayelitsha | 252 (236–265) | 58 (46–64) | 0.27 (0.11 to 0.42) | 0.002 | 26 (24–30) | 0.20 (0.03 to 0.36) | 0.019 | 2 (1–2) | 0.08 (−0.11 to 0.25) | 0.42 |
| South Africa, Themba Lethu | 171 (161–180) | 41 (37–44) | 0.22 (0.05 to 0.38) | 0.010 | 16 (14–17) | 0.15 (−0.03 to 0.31) | 0.097 | 1 (1–1) | 0.02 (−0.16 to 0.19) | 0.86 |
| South Africa, Tygerberg | 34 (34–37) | 7 (6–8) | 0.19 (0.02 to 0.35) | 0.028 | 3 (3–3) | 0.09 (−0.09 to 0.25) | 0.33 | 0.3 (0.2–0.5) | 0.11 (−0.06 to 0.28) | 0.22 |
| Zambia, CIDRZ | 2163 (2095–2233) | 275 (259–301) | 0.82 (0.75 to 0.87) | <0.001 | 9 (9–10) | 0.54 (0.40 to 0.65) | <0.001 | 29 (28–31) | 0.68 (0.57 to 0.77) | <0.001 |
| Zimbabwe, Newlands | 31 (29–34) | 5 (4–6) | 0.20 (0.02 to 0.37) | 0.029 | 2 (2–2) | 0.23 (0.06 to 0.40) | 0.011 | 0.2 (0.1–0.4) | 0.22 (0.05 to 0.39) | 0.014 |
| **All** | **170 (34–264)** | **40 (6–63)** | **0.28 (0.21 to 0.35)** | **<0.001** | **9 (3–17)** | **0.25 (0.18 to 0.32)** | **<0.001** | **1 (0–2)** | **0.14 (0.07 to 0.22)** | **<0.001** |

Median monthly event counts (2000–2014) and Pearson's pairwise correlation coefficients are shown.
ART, antiretroviral therapy; CIDRZ, Centre for Infectious Disease Research in Zambia; EPTB, extrapulmonary tuberculosis; KS, Kaposi sarcoma; PTB, pulmonary tuberculosis.

sites (figure 2). The strongest fluctuations were seen in Zimbabwe (Newlands), with PTB diagnosis peaks in January and troughs in December (+87% and −33% compared with the monthly average, respectively). Similar to Zimbabwe, all three South African sites showed marked drops in ART initiations and PTB diagnoses in December, in addition to a milder drop in April–May. In contrast, Zambia (CIDRZ) showed little variation in ART enrolments and PTB diagnoses throughout the year. We consistently observed a similar seasonal pattern in patients newly starting ART and PTB diagnoses every year (online supplementary figure 1), in men and women, as well as in younger and older patients (online supplementary figure 2).

The deviations of monthly event counts from the yearly averages confirmed the congruent fluctuations of ART initiation and PTB diagnosis (online supplementary figure 3). This was further supported by the correlation coefficients between the deviations of monthly ART initiations from yearly averages and the deviations of monthly PTB diagnoses from yearly averages (table 3): coefficient was 0.27 in Khayelitsha (95% CI 0.11 to 0.42, P=0.002), 0.22 in Themba Lethu (95% CI 0.05 to 0.38, P=0.010), 0.19 in Tygerberg (95% CI 0.02 to 0.35, P=0.028), 0.82 in CIDRZ (95% CI 0.75 to 0.87, P<0.001) and 0.20 in Newlands (95% CI 0.02 to 0.37, P=0.029). Regression coefficients between deviations of monthly PTB diagnosis counts and monthly ART initiation counts from yearly averages did not substantially change when adjusted for monthly temperatures and latitude (table 4). This was further supported by figure 3, which showed no evidence for a relationship between the adjusted regression coefficients and the latitude of the ART programme's location (r=0.43, P=0.47).

To determine whether the diagnosis of other OIs followed trends similar to that of PTB and ART initiations, we studied the monthly diagnoses of EPTB and KS, which we did not expect to be influenced by climatic factors. The

seasonal pattern of EPTB diagnoses paralleled that of ART initiation at most sites despite low numbers of EPTB diagnoses (1.8–26.5 mean monthly EPTB diagnoses). The deviations of monthly ART initiations from the yearly averages correlated with those of EPTB diagnoses in Khayelitsha, CIDRZ and Newlands (correlation coefficient range: 0.20–0.54, table 3). Although KS was a rare diagnosis averaging 1.1 cases per month overall, we observed that the fluctuations in KS diagnosis counts similarly followed changes in ART enrolment at all sites (table 3). As for PTB diagnoses, the adjusted regression models did not support any influence of temperature or latitude on the observed fluctuations in EPTB or KS diagnoses (table 4).

## DISCUSSION

We observed monthly variations in PTB diagnosis at ART programmes in South Africa, Zambia and Zimbabwe. These fluctuations occurred regardless of latitudes and followed trends in ART enrolment rather than seasonal variations in temperatures, hence mirroring fluctuations in clinical activity. In South Africa and Zimbabwe, a marked drop of ART enrolments and PTB diagnoses was seen in December, followed by a rebound in January–February. A milder drop was also observed in South African sites around April–May.

In Southern Africa, seasons are marked by cooler winters and warmer summers in Cape Town (Khayelitsha, Tygerberg), and become less pronounced as latitude decreases, in Johannesburg (Themba Lethu), Zimbabwe (Newlands) and Zambia (CIDRZ). Therefore, we expected to see stronger fluctuations in PTB diagnosis, with peaks following wintertime, in the Cape Town clinics situated at the Southern tip of South Africa, than in Johannesburg (Themba Lethu clinic), Zimbabwe (Newlands clinic) and Zambia (CIDRZ), where winter months are mild. Instead,

**Table 4**  Regression coefficients between deviations of monthly counts of PTB, EPTB or KS diagnoses and deviations of monthly counts of ART enrolments from the yearly averages ('unadjusted for temperature'); adjusted for the deviations of monthly air temperatures to yearly average air temperatures at the ART programme ('adjusted for temperature')

| ART programme | PTB diagnoses | | EPTB diagnoses | | KS diagnoses | |
|---|---|---|---|---|---|---|
| | Coefficient (95% CI) | P value | Coefficient (95% CI) | P value | Coefficient (95% CI) | P value |
| **South Africa, Khayelitsha** | | | | | | |
| Unadjusted for temperature | 0.33 (0.13 to 0.53) | 0.002 | 0.28 (0.05 to 0.51) | 0.019 | 0.30 (–0.43 to 1.03) | 0.416 |
| Adjusted for temperature | 0.30 (0.09 to 0.51) | 0.005 | 0.27 (0.03 to 0.51) | 0.027 | 0.44 (–0.31 to 1.12) | 0.248 |
| **South Africa, Themba Lethu** | | | | | | |
| Unadjusted for temperature | 0.16 (0.04 to 0.28) | 0.010 | 0.21 (–0.04 to 0.45) | 0.097 | 0.07 (–0.74 to 0.88) | 0.860 |
| Adjusted for temperature | 0.17 (0.05 to 0.29) | 0.005 | 0.21 (–0.03 to 0.46) | 0.089 | 0.10 (–0.71 to 0.92) | 0.805 |
| **South Africa, Tygerberg** | | | | | | |
| Unadjusted for temperature | 0.41 (0.04 to 0.78) | 0.028 | 0.20 (–0.20 to 0.60) | 0.328 | 0.82 (–0.49 to 2.14) | 0.218 |
| Adjusted for temperature | 0.41 (0.05 to 0.78) | 0.028 | 0.21 (–0.20 to 0.62) | 0.307 | 0.92 (–0.39 to 2.23) | 0.169 |
| **Zambia, CIDRZ** | | | | | | |
| Unadjusted for temperature | 0.84 (0.74 to 0.95) | <0.001 | 1.39 (1.00 to 1.78) | <0.001 | 0.91 (0.73 to 1.08) | <0.001 |
| Adjusted for temperature | 0.83 (0.72 to 0.94) | <0.001 | 1.40 (1.00 to 1.80) | <0.001 | 0.90 (0.72 to 1.08) | <0.001 |
| **Zimbabwe, Newlands** | | | | | | |
| Unadjusted for temperature | 0.29 (0.03 to 0.54) | 0.029 | 0.47 (0.11 to 0.83) | 0.011 | 1.32 (0.27 to 2.38) | 0.014 |
| Adjusted for temperature | 0.29 (0.037 to 0.55) | 0.025 | 0.47 (0.12 to 0.84) | 0.010 | 1.31 (0.26 to 2.36) | 0.015 |
| **All** | | | | | | |
| Unadjusted for temperature and latitude | 0.37 (0.27 to 0.47) | <0.001 | 0.50 (0.35 to 0.65) | <0.001 | 0.75 (0.35 to 1.16) | <0.001 |
| Adjusted for temperature | 0.38 (0.28 to 0.48) | <0.001 | 0.51 (0.36 to 0.66) | <0.001 | 0.77 (0.36 to 1.18) | <0.001 |
| Adjusted for temperature and latitude | 0.38 (0.28 to 0.48) | <0.001 | 0.51 (0.36 to 0.66) | <0.001 | 0.77 (0.36 to 1.18) | <0.001 |

The overall regression model was also adjusted for the latitude of the ART programme's location ('adjusted for temperature and latitude').
Monthly air temperatures are based on 1961–1990 averages.
ART, antiretroviral therapy; CIDRZ, Centre for Infectious Disease Research in Zambia; EPTB, extrapulmonary tuberculosis; KS, Kaposi sarcoma; PTB, pulmonary tuberculosis.

we observed variations in PTB that seemed independent from climatic changes. This might be explained by mild winters in Southern Africa, contrasting with countries with harsh winters. Mongolia typically has cold winters and strong seasonal patterns in TB notifications, likely due to increased airborne transmission from infectious PTB individuals resulting from indoor crowding and poorer ventilation during winter.[1 2] Nevertheless, seasonal variations in TB notifications in a population-based study of HIV-positive and HIV-negative individuals were documented in Cape Town, South Africa.[8]

We observed that variations in PTB diagnoses followed fluctuations in ART initiations regardless of temperatures and latitudes, and hence likely reflected activities at the clinic, rather than any climatic effects. This was supported by an association between changes in ART initiation and PTB diagnosis counts (unadjusted and adjusted for monthly climatic temperatures and latitude). Furthermore, important drops in ART initiations and PTB diagnoses were observed every year in December in South Africa and Zimbabwe, when clinical activities are reduced due to the holiday season. December drops were followed by peaks in January and February, when clinical activity resumed. Similar declines in PTB diagnoses at the end of the year have been reported in Zimbabwe and Uganda by Mabaera *et al.*[2] In CIDRZ, the largest of the cohorts, which we studied, monthly fluctuations in ART initiations and PTB diagnoses were the least marked, as compared with the South African and Zimbabwean sites. This can be explained by the shorter December holiday period in the Zambian health sector, in particular in the urban and peri-urban region of Lusaka, where CIDRZ is based. There, the ART programmes continue to run over the festive season in December and are only closing on official public holidays. Furthermore, HIV-positive individuals seeking care at CIDRZ are mainly permanent residents in Lusaka city (as opposed to migrants from rural areas), and hence less likely to leave the city to visit their family in other parts of the country during holidays, which would contribute to delayed TB diagnoses. Similarly, Easter vacations likely explain the milder drop observed in April–May in South Africa, and to a lesser extend in Zimbabwe, but not in Zambia. Age and gender did not seem to influence the seasonal variations.

EPTB and KS diagnoses also followed variations in ART initiations in our study. We did not expect the development of EPTB to be strongly marked by climatic changes. Indeed, EPTB is less likely resulting from recent transmission than PTB, the transmission mode of which is airborne, and hence more likely to be influenced by indoor crowding and reduced ventilation during winter.[1 7] Seasonal changes in EPTB case detection have been reported.[3 4 21] However, the underlying reasons for EPTB seasonality may considerably differ from those involved in PTB. In addition, EPTB is often underdiagnosed due to lack of adequate laboratory diagnostic tools in under-resourced settings.[3 21] Similarly to EPTB, KS was not expected to be influenced by climatic factors. The observed seasonal fluctuations of EPTB and KS diagnoses hence reflect variations in clinical activities, such as observed for PTB.

Several explanations of why PTB diagnoses may vary seasonally have been offered.[1 3 5 7–9] Beyond indoor crowding[1 5 9] and reduced clinical activities at specific times during the year,[2] health-seeking behaviour is also influencing access to care. Health-seeking behaviour might reflect environmental barriers (long distances to the clinic and waiting time),[22 23] psychological barriers (stigma)[24] and/or clinical barriers (poor patient's health status and disabilities).[25] Access to clinics can be problematic when temperatures are low and weather conditions are bad, which is more likely to be the case in places with harsh winters. Under these circumstances, PTB diagnosis might

be delayed to spring. Professional activities or beliefs may also prevent early healthcare seeking. Economical aspects play an important role, where care is to be fully or partly paid by patients.[15] These aspects might differently affect individuals, depending on gender or age.[26–29] However, in our study populations, trends in ART initiations and PTB diagnoses were independent from gender or age. Finally, the role of vitamin D deficiency has been debated, and studies of the relationship between vitamin D deficiency during winter and PTB are inconsistent.[3 30–32]

Our study is limited by the possible underascertainment of TB diagnoses at ART programmes and by the fact that we could only include IeDEA-SA sites that routinely collect information on OIs. However, the included ART programmes cover a wide range of geographical latitudes and different climatic seasons, with Khayelitsha and Tygerberg at the Southern tip and CIDRZ at the most equatorial end of the Southern African region. The findings of our study is restricted to HIV-positive populations, since trends in TB diagnosis and other OIs at ART programmes may differ from trends in HIV-negative populations. Due the high early TB mortality, HIV-positive individuals may not reach clinical care for a TB diagnosis.[33] In addition, although ART reduces the risk of OIs significantly, people living with HIV remain at a higher risk of developing TB.[34 35] Finally, social and environmental factors, including housing conditions, distance to healthcare facilities, access to care, disease awareness and health-seeking behaviour, may differently

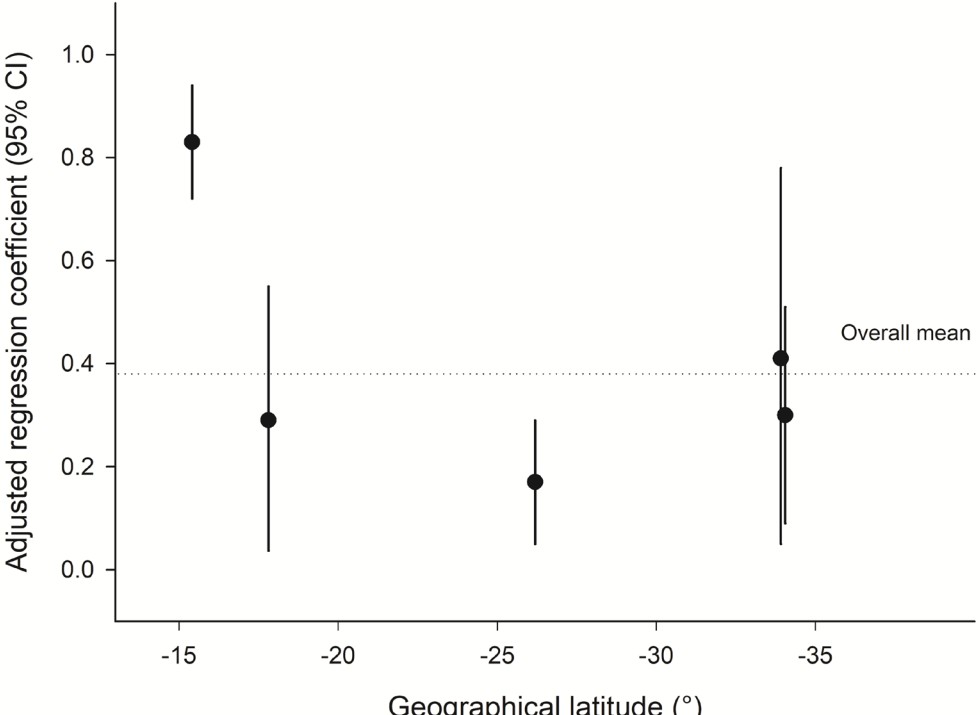

**Figure 3** Relationship between PTB diagnosis and geographical latitude: regression coefficients between deviations of monthly counts of PTB diagnoses and deviations of monthly counts of ART enrolments from the yearly averages (adjusted for air temperature) according to the geographical latitude of the ART programme's location. ART, antiretroviral therapy ; PTB, pulmonary tuberculosis.

affect the timely diagnosis of TB in HIV-positive compared with HIV-negative populations.[1 7 9 36 37] Unfortunately, this information is not routinely collected at ART programmes in the IeDEA-SA collaboration.

## CONCLUSIONS

In conclusion, our study suggests that yearly trends in PTB diagnosis at ART programmes in Southern Africa are mainly driven by seasonal variations in clinical activities, access to care and health-seeking behaviour, rather than climatic seasons. Our results underline the importance of offering regular care for HIV-infected individuals, including screenings for OIs such as TB at ART initiation and during follow-up visits, as part of integrated TB/HIV clinical activities at ART programmes.[38–41]

**Author affiliations**
[1]Institute of Social and Preventive Medicine, University of Bern, Bern, BE, Switzerland
[2]Division of Infection Diseases, University of Alabama at Birmingham, Birmingham, Alabama, USA
[3]Tuberculosis Department Unit, Centre for Infectious Disease Research in Zambia (CIDRZ), Lusaka, Zambia
[4]Centre for Infectious Disease Epidemiology and Research (CIDER), School of Public Health and Family Medicine, University of Cape Town, Cape Town, South Africa
[5]Médecins Sans Frontières, Khayelitsha, South Africa
[6]Departments of Epidemiology and Global Health, Boston University, Boston, USA
[7]Department of Internal Medicine, Health Economics and Epidemiology Research Office, School of Clinical Medicine, Faculty of Health Sciences, University of the Witwatersrand, Johannesburg, South Africa
[8]Division of Infectious Diseases, Department of Medicine, University of Stellenbosch & Tygerberg Academic Hospital, Cape Town, South Africa
[9]Newlands Clinic, Harare, Zimbabwe

**Acknowledgements** The authors thank all participating sites and patients whose data were used in this study. They also would like to thank collaborators who contributed to recording and entering data, as well as the data centres of the IeDEA-SA collaboration for preparing the data. They thank Christopher Ritter for editorial services.

**Collaborators** Matthias Egger (co-PI), University of Bern, Switzerland; Mary-Ann Davies (co-PI), University of Cape Town, South Africa; Frank Tanser, Africa Centre for Health and Population Studies, University of Kwazulu-Natal, South Africa; Michael Vinikoor, Centre for Infectious Disease Research in Zambia; Eusebio Macete, Centro de Investigação em Saúde de Manhiça, Mozambique; Robin Wood, Desmond Tutu HIV Centre (Gugulethu and Masiphumelele clinics), South Africa; Kathryn Stinson, Khayelitsha ART Programme and Médecins Sans Frontières, South Africa; Geoffrey Fatti, Kheth'Impilo Programme, South Africa; Sam Phiri, Lighthouse Trust Clinic, Malawi; Cleophas Chimbetete, Newlands Clinic, Zimbabwe; Kennedy Malisita, Queen Elizabeth Hospital, Malawi; Brian Eley, Red Cross War Memorial Children's Hospital and Department of Paediatrics and Child Health, University of Cape Town, South Africa; Jochen Ehmer, Solidarmed, Switzerland; Christiane Fritz, SolidarMed SMART Programme, Lesotho; Michael Hobbins, SolidarMed SMART Programme, Mozambique; Kamelia Kamenova, SolidarMed SMART Programme, Zimbabwe; Matthew Fox, Themba Lethu Clinic, South Africa; Hans Prozesky, Tygerberg Academic Hospital, South Africa; Karl Technau, Empilweni Clinic, Rahima Moosa Mother and Child Hospital, South Africa; Shobna Sawry, Harriet Shezi Children's Clinic, Chris Hani Baragwanath Academic Hospital, South Africa.

**Contributors** MB: analysed the data and completed the final draft of the manuscript. LF: conceptualised the project. MZ: established the methodology. KZ, SER, AB, MPF, HWP, CC and ME: provided input into the study design, analyses and drafting of the paper. All authors: reviewed and approved the final version of the manuscript.

**Funding** Research reported in this publication was supported by the National Institute Of Allergy And Infectious Diseases of the National Institutes of Health under Award Number U01AI069924. The content is solely the responsibility of the authors

and does not necessarily represent the official views of the National Institutes of Health. MPF was supported by USAID 674-A-12-00029 from the United States Agency for International Development. ME was supported by special project funding (Grant No. 174281) from the Swiss National Science Foundation.

**Disclaimer** The opinions expressed herein are those of the authors and do not necessarily reflect the views of the funders. The funders had no role in study design, data collection and analysis, decision to publish or preparation of the manuscript.

**Competing interests** None declared.

**Ethics approval** This project was approved by the Cantonal Ethics Committee Bern (swissethics, Bern, Switzerland) and the University of Cape Town Human Research Ethics Committee (Cape Town, South Africa).

**Provenance and peer review** Not commissioned; externally peer reviewed.

**Data sharing statement** The individual contributing cohorts own the data. Consortium members may request data for analysis according to standard procedures. The consortium can also accept concept proposals from qualified researchers external to the collaboration. For more information, see www.iedea.org/working-groups/executive-committee.

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
