## [Reviewer comments · BMJ Open]

ARTICLE DETAILS

TITLE (PROVISIONAL)	Seasonal variations in tuberculosis diagnosis among HIV-positive individuals in Southern Africa: analysis of cohort studies at antiretroviral treatment programs
AUTHORS	Ballif, Marie; Zürcher, Kathrin; Reid, Stewart; Boule, Andrew; Fox, Matt; Prozesky, Hans; Chimbete, Cleophas; Zwahlen, Marcel; Egger, Matthias; Fenner, Lukas

VERSION 1 – REVIEW

REVIEWER	Worodria William Department of Medicine, Makerere University, Kampala, Uganda
REVIEW RETURNED	08-Jun-2017

GENERAL COMMENTS	The study investigated whether the variations in the trend in pulmonary tuberculosis (PTB) diagnosis from five ART programmes in Southern Africa was associated with seasonal variations. This is an important study which is very well written and provides useful information. The group has a large database from various ART providing sites which has potential to answer important clinical and programmatic questions. One would have wanted to have additional information/ justification for the following: 1. It is well known that certain viral and bacterial infections are worse during the winter months but for diseases like TB, with a relatively long and variable latency period, seasonal variation could be fairly difficult to study without addressing socioeconomic issues, housing, immune-related issues , etc.2. TB treatment unlike ART initiation is usually started immediately after diagnosis for public health reasons. ART on the other hand is dependent on how the programmes are organized. I am not sure analysis of trends of ART initiation would provide any seasonal data.3. One would want to have a justification of why PTB is considered different from EPTB, especially that EPTB is a commoner presentation in HIV co-infected individuals and frequently co-exists with PTB. I do not think it should be considered another OI. It is a different manifestation of the same TB. If PTB is suspected to be seasonal why EPTB should not also be considered the same?4. It would be helpful to describe how the HIV and TB care in the three countries considered varied regarding funding, guideline
--

	recommendation for ART initiation, whether or not they were routinely given co-trimoxazole prophylaxis. In addition what is the cost to the subjects accessing care? 5. The article is lacking in data collected regarding important confounders like housing conditions, socioeconomic status, and location of the care facilities –whether rural or urban. 6. It is important to note that these clinics are located in ART providing sites and the level of integration or collaboration of HIV and TB services has been a challenge. Does this present a true picture of the TB burden or an underestimate? Is this a source of potential bias? Noted one typo ...line 178 with the abbreviation EPTB
--	---

REVIEWER	Keren Middelkoop University of Cape Town, South Africa I have previously collaborated with some of the authors on this paper. However, to confirm: I have not been involved in this analysis or manuscript
REVIEW RETURNED	14-Jun-2017

GENERAL COMMENTS	This is an observational cohort study that assessed for seasonal variation in PTB rates across 5 sites in 3 countries. The hypothesis the authors aimed to assess was whether “seasonal climate fluctuations in TB diagnoses would likely be stronger in countries located farther away from the equator”. Variation in monthly PTB rates were plotted against ART initiation rates at the same sites. The study findings include that variations in PTB rates differed across sites, at seemed to be more closely related to clinic activity than seasonal factors. This is a well written and succinct paper, on a topic of interest. This study was a secondary analysis performed on pre-existing databases: It is hard to comment on the suitability of the databases used beyond noting that while there is some variability in latitude, there were no equatorial countries included, which may have reduced the ability to detect a significant trend. In addition, restricting analysis to patients attending ART clinics has notable limitations. The analysis had some limitations, note under major comments. The conclusions of the study are justified, and the authors clearly limit their interpretations and suggestions to HIV-positive patients on ART, as appropriate. Minor comments While the authors do note the limitation of restricting analysis to patients attending ART clinics a more indepth discussion of how they think this may have affected the results would improve the paper. Major comments: The hypothesis stated that “seasonal climate fluctuations in TB diagnoses would likely be stronger in countries located farther away
---

	from the equator". However, the analysis does not include the latitude of the sites. The analysis does incorporate temperature fluctuations as a proxy, but it seems that latitude would be an easy factor to assess, and I am not clear why the authors did not do this.
--	---

REVIEWER	James Shepherd Yale School of Medicine No Competing Interest
REVIEW RETURNED	12-Jul-2017

GENERAL COMMENTS	The manuscript by Balliff et al investigates the link between southern African seasons and the incidence of pulmonary TB. The methodology uses a large database of clinical records from a network of clinics in South Africa, Zimbabwe and Zambia to link the month with the number of new TB diagnoses recorded and normalize to the volume of patient visits using regression analysis by ART initiations. The initial hypothesis was that an increase in winter diagnoses of TB might be seen with magnitude greater in the south in accordance with some earlier studies in the northern hemisphere. The presumption is that during the chilly winter months there may be greater indoor crowding with increased transmission of TB. As a control the monthly diagnoses of extrapulmonary TB and Kaposi's Sarcoma were used which were not expected to fluctuate seasonally. The results describe no increase in TB diagnoses during the winter months and show a very close relationship between initiations of ART and TB diagnoses. In fact all three diagnoses are proportional to the number of patients initiating ART. There is a seasonal dip in all metrics during December, mid summer, which is presumed to be due to the long holiday customarily taken in Africa with decreased staffing and closure of the ART clinics. The most striking finding presented is the relative lack of decrease in diagnoses and ART initiations at CIDRZ in Zambia during December. The report is well written and the subject important. The data methods appear appropriate and the numbers included are very large from the leDEA-SA database which links the sites. I have the following observations and questions;  1. It is not clear how the data in the leDEA database is collected and what variables are recorded. In particular the authors report using the WHO Guidelines 2011 for diagnosing TB which recommends sputum examination, CXR and clinical exam but the suspicion, supported by many other studies, is that most TB diagnosis in Africa is empiric. This is backed up by the close correlation between visits and TB diagnoses. Therefore the study is not able to make any link between TB transmission and seasonality. 2. It might be expected that TB transmission occurs throughout the year even with seasonal variations in magnitude of transmission and that the time between infection and frank pulmonary TB is also variable, depending on other clinical factors, in particular CD4 count in PLHIV. Therefore is it reasonable to expect to see any seasonal variation using the reports' methodology?
---

	3. The most striking finding is the marked drop in all metrics during December in South Africa and Zimbabwe but not in Zambia. This is an important public health finding, especially since there appears to be a 'jump' in diagnoses in January which suggests a delay in TB management. What is it about CIDRZ that avoids this administrative problem? Could it be replicated in the other sites? What is the public health implication based on the magnitude of 'drop' and 'rebound' observed? Could this be modeled? In conclusion, I do not think the data available and methods used are capable of answering seasonal changes in confirmed pulmonary TB. I, and many public health policy makers, would be very interested to see more analysis and discussion of the main finding of the study – the differences between the sites in December.
--	---

VERSION 1 – AUTHOR RESPONSE

Reviewer 1

The study investigated whether the variations in the trend in pulmonary tuberculosis (PTB) diagnosis from five ART programmes in Southern Africa was associated with seasonal variations.

This is an important study which is very well written and provides useful information.

The group has a large database from various ART providing sites which has potential to answer important clinical and programmatic questions.

Authors' response: Thank you.

One would have wanted to have additional information/ justification for the following:

1. It is well known that certain viral and bacterial infections are worse during the winter months but for diseases like TB, with a relatively long and variable latency period, seasonal variation could be fairly difficult to study without addressing socioeconomic issues, housing, immune-related issues, etc.

Authors' response: We agree that social information such as socioeconomic status and housing are important factors in the development of PTB. Unfortunately, this information is not available in the routinely collected dataset in the leDEA-SA collaboration and would be difficult to obtain retrospectively. However, we discussed the importance of these aspects in the Discussion section of the revised manuscript (p. 11). We also discuss the lack of socioeconomic information in the limitation paragraph (p. 12).

2. TB treatment unlike ART initiation is usually started immediately after diagnosis for public health reasons. ART on the other hand is dependent on how the programmes are organized. I am not sure analysis of trends of ART initiation would provide any seasonal data.

Authors' response: We agree with the reviewer that the initiation of ART may depend on the programmes' management and administration structure, as well as on local habits, local economical situation, and patients' behaviour. These factors are reflected in the seasonal (but not necessarily climatic) patterns in ART initiation at all sites, with drops in TB diagnoses likely due to changes in the intensity of clinical activity. However, we realized that the distinction between seasonal fluctuations (descriptive term for changes over a one-year period) and climatic fluctuations (temperature changes

over a one-year period) was not always clear to the reader. We therefore clarified this throughout the manuscript to avoid any confusions (p. 6, 10 and 11).

3. One would want to have a justification of why PTB is considered different from EPTB, especially that EPTB is a commoner presentation in HIV co-infected individuals and frequently co-exists with PTB. I do not think it should be considered another OI. It is a different manifestation of the same TB. If PTB is suspected to be seasonal why EPTB should not also be considered the same?

Authors' response: We agree with the reviewer that EPTB is a more common clinical presentation in HIV-positive compared to HIV-negative patients. In our study, we mainly focused on PTB cases separately from EPTB cases, as PTB is the best available indicator for recent TB transmission (only childhood TB would be a more appealing indicator), which can be influenced by seasonal changes in time spent indoors. Furthermore, due to the difficulties to diagnose EPTB in underresourced settings, the proportion of undiagnosed EPTB cases is likely to be higher than in PTB. In agreement with the reviewer, we do not consider EPTB as another OI. The definitions of PTB and EPTB are presented in the manuscript (p. 6), and we expanded the discussion on EPTB in the Discussion section (p. 11) to further clarify these issues.

4. It would be helpful to describe how the HIV and TB care in the three countries considered varied regarding funding, guideline recommendation for ART initiation, whether or not they were routinely given co-trimoxazole prophylaxis. In addition what is the cost to the subjects accessing care?

Authors' response: Thank you for this comment. We present the requested information in a new table according to the reviewer's suggestion (Table 1, p. 19).

"Table 1: Characteristics of participating ART programs."

Characteristics South Africa – Khayelitsha South Africa – Themba Lethu South Africa – Tygerberg
Zambia – CIDRZ
Zimbabwe – Newlands

Geographical latitude (°) -34.05 -33.91 -26.12 -15.41 -17.81

Setting Urban Urban Urban Combination of urban and peri-urban health centres Urban

Level of care Primary Secondary Tertiary Primary Primary

TB case definition Majority microbiologically confirmed, others based on clinical presentation

Microbiologically confirmed Majority microbiologically confirmed, others based on clinical presentation

Majority microbiologically confirmed, others based on clinical presentation Majority microbiologically confirmed, others based on clinical presentation

Cost of TB diagnosis to patients Available at no cost Available at no cost Partial payment Available at no cost or partial payment (depending on facility) Available at no cost

Co-trimoxazole for prevention All patients According to CD4 cell count According to CD4 cell count

According to CD4 cell count or to all patients (depending on facility) According to CD4 cell count

Integration of HIV/TB clinical services Same facility / same day appointments Same facility / same day appointments Cross referral Cross referral Same facility / same day appointments

TB screening at ART enrolment All patients, symptomatic screening and diagnostic tests All patients, symptomatic screening only All patients, symptomatic screening only All patients, symptomatic screening and diagnostic tests at selected sites All patients, symptomatic screening only.

5. The article is lacking in data collected regarding important confounders like housing conditions, socioeconomic status, and location of the care facilities –whether rural or urban.

Authors' response: We agree with the reviewer that housing conditions and socioeconomic factors might influence the risk of recent transmission, health seeking behaviour, and fluctuations in clinical activity throughout the year. Unfortunately, this information is not routinely collected at ART programs participating in leDEA-SA, and would be difficult to collect retrospectively. This has been acknowledged as a limitation (p. 12, see also response to point 2 above).

We also agree that site-specific characteristics of the participating ART programs are important (e.g., urban versus rural setting). Therefore, we added the relevant site-specific characteristics in a new table (Table 1, p. 19; see also response to point 4 above).

6. It is important to note that these clinics are located in ART providing sites and the level of integration or collaboration of HIV and TB services has been a challenge. Does this present a true picture of the TB burden or an underestimate? Is this a source of potential bias?

Authors' response: The management of TB in HIV-coinfected individuals in care at ART programs might not be representative of the management of TB in HIV-negative individuals. This is a limitation that we acknowledge (p. 12, see also responses to reviewer 2). In addition, TB (and particularly EPTB) are generally conditions that remain underdiagnosed, especially in people living with HIV due to the paucibacillary nature of the disease. In addition, diagnostic capacities at ART programs in sub-Saharan Africa are limited (see Fenner et al., PLOS ONE 2013; Charles et al., PLOS ONE, 2016), which can lead to the underestimation of TB diagnoses. However, we do not think that this will affect the seasonal fluctuations and the comparison of seasonal changes across different ART programs.

7. Noted one typo ...line 178 with the abbreviation EPTB

Authors' response: Thank you. This was corrected (p. 9).

Reviewer 2

Comment: This is an observational cohort study that assessed for seasonal variation in PTB rates across 5 sites in 3 countries. The hypothesis the authors aimed to assess was whether "seasonal climate fluctuations in TB diagnoses would likely be stronger in countries located farther away from the equator". Variation in monthly PTB rates were plotted against ART initiation rates at the same sites. The study findings include that variations in PTB rates differed across sites, at seemed to be more closely related to clinic activity than seasonal factors.

This is a well written and succinct paper, on a topic of interest.

Authors' response: Thank you.

Comment: This study was a secondary analysis performed on pre-existing databases: It is hard to comment on the suitability of the databases used beyond noting that while there is some variability in latitude, there were no equatorial countries included, which may have reduced the ability to detect a significant trend. In addition, restricting analysis to patients attending ART clinics has notable limitations.

Authors' response: We agree that data from ART programs closer to the equator would be a benefit to this analysis. However, our study was limited by sites participating in the leDEA-SA collaboration; among those, we selected ART programs covering the range of possible latitudes. We acknowledged this limitation in the Discussion section (p. 12).

The issue of generalisability to TB patients in the general population, particularly the HIV-negative TB population, has also been addressed in the Discussion section (p. 12). Please see also response to reviewer 1 (point 6) and response to point 1 below.

Comment: The analysis had some limitations, note under major comments.

The conclusions of the study are justified, and the authors clearly limit their interpretations and suggestions to HIV-positive patients on ART, as appropriate.

Authors' response: Thank you.

1. Minor comments

While the authors do note the limitation of restricting analysis to patients attending ART clinics, a more in-depth discussion of how they think this may have affected the results would improve the paper.

Authors' response: As mentioned above, we acknowledge that seasonal variations in TB diagnoses in HIV-positive patients at ART programs might differ from trends in the general population. As the majority of the literature focuses on the general population, the aim of this article was to describe seasonal variations in TB diagnoses at ART programs. We agree with the reviewer that our study population is different from the general population, and therefore we expanded the text on this topic in the Discussion section as follows (p. 12):

... "The findings of our study is restricted to HIV-positive populations, since trends in TB diagnosis and other OIs at ART programs may differ from trends in HIV-negative populations. Although ART reduces the risk of OIs significantly, people living with HIV remain at a higher risk of developing TB. In addition, due the high early TB mortality, HIV-positive individuals may not reach clinical care for a TB diagnosis. Finally, social and environmental factors, including housing conditions, distance to health care facilities, access to care, disease awareness, and health-seeking behaviour may differently affect the timely diagnosis of TB in HIV-positive compared to HIV-negative populations. Unfortunately, this information is not routinely collected at ART programs in the leDEA-SA collaboration." ...

2. Major comments

The hypothesis stated that "seasonal climate fluctuations in TB diagnoses would likely be stronger in countries located farther away from the equator".

However, the analysis does not include the latitude of the sites. The analysis does incorporate temperature fluctuations as a proxy, but it seems that latitude would be an easy factor to assess, and I am not clear why the authors did not do this.

Authors' response: Thank you for this valuable comment. As each of the participating site presents a different geographical latitude and all analyses were presented stratified by ART program, we accounted for the geographical distance from the equator in relation to seasonal fluctuations. However, we agree with the reviewer that the geographical latitude component should be better reflected in the analyses. Therefore, we made the following modifications in the manuscript according to the reviewer's suggestion:

- We incorporated the geographical coordinates of each of the participating ART programs in Table 1 on site characteristics (new table, p. 19).

- In Table 4, which shows the regression coefficients between deviations of monthly counts of PTB diagnoses and deviations of monthly counts of antiretroviral therapy enrolments from the yearly averages, we also present the estimates adjusted for the air temperatures as well as the geographical latitude of the ART program (p. 22). The regression coefficients did not substantially change when

adjusted for monthly temperatures and latitudes. These results are highlighted in the Results (p. 9) and in the Discussion section (p. 10).

- Finally, we also produced a figure plotting the regression coefficients (adjusted for monthly air temperatures) according to the geographical latitude of the ART program's location (Figure 3, p. 26): "Figure 3: Relationship between pulmonary tuberculosis (PTB) diagnosis and geographical latitude: regression coefficients between deviations of monthly counts of PTB diagnoses and deviations of monthly counts of antiretroviral therapy enrolments from the yearly averages (adjusted for air temperature) according to the geographical latitude of the ART program's location."

Reviewer 3

The manuscript by Ballif et al investigates the link between southern African seasons and the incidence of pulmonary TB. The methodology uses a large database of clinical records from a network of clinics in South Africa, Zimbabwe and Zambia to link the month with the number of new TB diagnoses recorded and normalize to the volume of patient visits using regression analysis by ART initiations. The initial hypothesis was that an increase in winter diagnoses of TB might be seen with magnitude greater in the south in accordance with some earlier studies in the northern hemisphere. The presumption is that during the chilly winter months there may be greater indoor crowding with increased transmission of TB. As a control the monthly diagnoses of extrapulmonary TB and Kaposi's Sarcoma were used which were not expected to fluctuate seasonally.

The results describe no increase in TB diagnoses during the winter months and show a very close relationship between initiations of ART and TB diagnoses. In fact all three diagnoses are proportional to the number of patients initiating ART. There is a seasonal dip in all metrics during December, mid summer, which is presumed to be due to the long holiday customarily taken in Africa with decreased staffing and closure of the ART clinics. The most striking finding presented is the relative lack of decrease in diagnoses and ART initiations at CIDRZ in Zambia during December.

The report is well written and the subject important. The data methods appear appropriate and the numbers included are very large from the leDEA-SA database which links the sites.

Authors' response: Thank you.

I have the following observations and questions;

1. It is not clear how the data in the leDEA database is collected and what variables are recorded. In particular the authors report using the WHO Guidelines 2011 for diagnosing TB which recommends sputum examination, CXR and clinical exam but the suspicion, supported by many other studies, is that most TB diagnosis in Africa is empiric. This is backed up by the close correlation between visits and TB diagnoses. Therefore the study is not able to make any link between TB transmission and seasonality.

Authors' response: As the reviewer correctly points out, a considerable proportion of TB diagnoses relies on clinical criteria and remains bacteriologically unconfirmed in underresourced settings, such as in Southern Africa, where our study sites are located. We would like to clarify, that, at the leDEA-SA sites, TB cases include both bacteriologically confirmed and clinically diagnosed TB (according to the revised WHO definitions, WHO 2013). Although we can only estimate the exact proportion of clinically diagnosed (or empiric) TB cases, it is reasonable to estimate that not all TB cases could be bacteriologically confirmed. We made the following changes in the manuscript to clarify this issue:

- In the new Table 1, we present the site-specific characteristics including the case definition of TB (p. 19).
- We clarified the sentence on the TB case definition in the Methods section (p. 6):
... "PTB was defined according to the case definition in use at the participating leDEA-SA sites (Table 1)" ...

2. It might be expected that TB transmission occurs throughout the year even with seasonal variations in magnitude of transmission and that the time between infection and frank pulmonary TB is also variable, depending on other clinical factors, in particular CD4 count in PLHIV. Therefore is it reasonable to expect to see any seasonal variation using the reports' methodology?

Authors' response: Seasonal variations in PTB diagnosis in relationship with temperature and climatic variations throughout the year has been reported in the general population, and associated with different social behaviour and time spent indoors favouring transmission in winter. The most extreme example was describe in Mongolia, with its pronounced continental climate (Naranbat et al., Eur Respir J, 2009, Discussion section, p. 12). Based on these observations, we investigated whether similar climatic seasonal trends were seen in ART programs in Southern Africa. Our analyses indeed showed seasonal variations in the number of PTB diagnosis, but these variations were paralleling the number of ART enrolments (as an indicator of clinical activities), rather than climatic (or latitude) variations.

3. The most striking finding is the marked drop in all metrics during December in South Africa and Zimbabwe but not in Zambia. This is an important public health finding, especially since there appears to be a 'jump' in diagnoses in January which suggests a delay in TB management. What is it about CIDRZ that avoids this administrative problem? Could it be replicated in the other sites? What is the public health implication based on the magnitude of 'drop' and 'rebound' observed? Could this be modeled?

Authors' response: The milder fluctuations in PTB diagnosis observed in Zambia (CIDRZ) can be explained by the absence of an extended holiday period at the ART programs, which only close on the official public holidays in December. In contrast, other ART programs are closing for up to several weeks, during which clinical staff are on leave and many of the patients visit their family and relatives elsewhere. Therefore, TB diagnosis is delayed to January, when they return and when clinical activities resume. These aspects are now outlined in details in the Discussion section as follows (p. 11):

... "This can be explained by the shorter December holiday period in the Zambian health sector, in particular in the urban and peri-urban region of Lusaka, where CIDRZ is based. There, the ART programs continue to run over the festive season in December and are only closing on official public holidays. Likewise, most HIV-positive individuals seeking care at CIDRZ are permanent resident in Lusaka city (as opposed to being migrants from the rural areas), and hence less likely to leave the city to visit their family in other parts of the country further away, which would contribute to delay TB diagnosis." ...

We agree with the reviewer that modelling work to estimate the impact of these drops and rebounds of clinical activities on the health care system would be interesting, but such work would be beyond the scope of the current manuscript.

In conclusion, I do not think the data available and methods used are capable of answering seasonal changes in confirmed pulmonary TB. I, and many public health policy makers, would be very interested to see more analysis and discussion of the main finding of the study – the differences between the sites in December.

Authors' response: We would like to emphasize again that we analysed all TB cases (bacteriologically confirmed and clinically diagnosed), and not just confirmed PTB. Please see response to point 1 above for further details.

Regarding the absence of December drop observed in CIDRZ, we highlighted in the Discussion section the potential impact of extended holiday periods, when clinics reduce their activity. Please see response to point 3 above for further details.

VERSION 2 – REVIEW

REVIEWER	Worodria William Mulago Hospital, Kampala, Uganda
REVIEW RETURNED	25-Aug-2017

GENERAL COMMENTS	I agree with the revisions made to the manuscript.
--

REVIEWER	Keren Middelkoop University of Cape Town, South Africa I have previously collaborated with some of the authors on this paper. However, I have not been involved in this analysis or manuscript
REVIEW RETURNED	08-Sep-2017

GENERAL COMMENTS	No further comments
---------------------

REVIEWER	James Shepherd Yale University School of Medicine USA
REVIEW RETURNED	30-Sep-2017

GENERAL COMMENTS	Although my reservations about the ability of the study approach to detect changes in the TRANSMISSION of TB throughout the year have been somewhat addressed, mainly through referring to other studies in the bibliography and a brief recognition of the difference between HIV negative and HIV positive TB dynamics, my scepticism remains. Nevertheless it is a well written and interesting study, particularly for anyone with familiarity with the large HIV treatment programs in southern Africa and for me the increased discussion of 'holiday diagnostic variations' is very interesting. One last point - to my knowledge there is still a very defined winter season in Johannesburg and Harare where the populations are expected to spend more time in close contact indoors and so the described latitudinal gradation of a winter season influencing human behavior may not apply very strongly.
--

VERSION 2 – AUTHOR RESPONSE

Minor comment from Reviewer 3

Although my reservations about the ability of the study approach to detect changes in the TRANSMISSION of TB throughout the year have been somewhat addressed, mainly through referring to other studies in the bibliography and a brief recognition of the difference between HIV negative and HIV positive TB dynamics, my scepticism remains. Nevertheless it is a well written and interesting study, particularly for anyone with familiarity with the large HIV treatment programs in southern Africa and for me the increased discussion of 'holiday diagnostic variations' is very interesting.

Response from the authors: Thank you.

Comment: One last point - to my knowledge there is still a very defined winter season in Johannesburg and Harare where the populations are expected to spend more time in close contact indoors and so the described latitudinal gradation of a winter season influencing human behavior may not apply very strongly.

Response from the authors: Thank you. We agree with the reviewer that variations in the air temperature can be observed in Johannesburg and Harare, with the months of November/December being somewhat colder. Cape Town at the Southern tip of South Africa certainly has more pronounced climatic seasons. To account for climatic differences across sites, we adjusted our models for geographical latitudes as well as for average air temperatures at the ART programs when assessing fluctuations in PTB diagnosis throughout calendar years. This is shown in Table 4 (Regression coefficients between deviations of monthly counts of PTB and deviations of monthly counts of ART enrolments from the yearly averages, adjusted for monthly air temperatures and latitude of the ART program's location) and in Figure 3 (Relationship between PTB diagnosis and geographical latitude). Therefore, we believe that we have addressed the concerns raised by the Reviewer. To make these points clearer, we rephrased some sentences in the discussion section of the manuscript (pages 10, 11 and 12).